# Phylogenetic trees of closely related bacterial species and subspecies based on frequencies of short nucleotide sequences

Yoshio Nakano[1]*, Yusaku Domon[2], Kenji Yamagishi[2]

**1** Department of Chemistry, Nihon University School of Dentistry, Tokyo, Japan, **2** Department of Chemical Biology and Applied Chemistry, Nihon University, College of Engineering, Koriyama, Fukushima, Japan

* nakano.yoshio70@nihon-u.ac.jp

**Data Availability Statement:** All bacterial genome sequence files are available from the NCBI Microbial Genomes (ftp://ftp.ncbi.nlm.nih.gov/genomes/Bacteria/) database and the chloroplast

## Abstract

Bacterial phylogenetic analyses are commonly performed to explore the evolutionary relationships among various bacterial species and genera based on their 16S rRNA gene sequences; however, these results are limited by mosaicism, intragenomic heterogeneity, and difficulties in distinguishing between related species. In this study, we aimed to perform genome-wide comparisons of different bacterial species, namely *Escherichia coli*, *Shigella*, *Yersinia*, *Klebsiella*, and *Neisseria* spp., based on their K-mer profiles to construct phylogenetic trees. Pentanucleotide frequency analyses (512 patterns of 5 nucleotides each) were performed to distinguish between highly similar species. Moreover, *Escherichia albertii* strains were clearly distinguished from *E. coli* and *Shigella*, despite being closely related to enterohemorrhagic *E. coli* in the phylogenetic tree. In addition, our phylogenetic tree of *Ipomoea* species based on pentamer frequency in chloroplast genomes was correlated with previously reported morphological similarities. Furthermore, a support vector machine clearly classified *E. coli* and *Shigella* genomes based on their pentanucleotide profiles. These results suggest that phylogenetic analyses based on penta- or hexamer profiles are a useful methodology for microbial phylogenetic studies. In addition, we introduced an R application, Phy5, which generates a phylogenetic tree based on genome-wide comparisons of pentamer profiles. The online version of Phy5 can be accessed at https://phy5.shinyapps.io/Phy5R/ and its command line version Phy5cli can be downloaded at https://github.com/YoshioNakano2021/phy5.

## Introduction

Currently, prokaryotic phylogenetic classification depends on 16S rRNA gene sequences, which are ubiquitously present and highly conserved in bacteria, but species with more than ≥ 99% identity based on 16S rDNA sequencing are rarely classified. In such cases, one or more additional conserved genes are often used as a secondary candidate for indexing during phylogenetic analyses, such as *gyrB* for *Shigella*, *Salmonella*, and *E. coli* [1]; *trpA*, *trpB*, *pabB*, and *putP* for *Shigella*, *Salmonella*, and *E. coli* [2]; and *thrA*, *trpE*, *glnA*, *tmk*, and *dmsA* for

genomes of Ipomoea species were obtained from the ftp site at NCBI 64 plastid sequences database (https://ftp.ncbi.nlm.nih.gov/refseq/release/plastid/).

**Funding:** This work was supported by JSPS KAKENHI Grant Numbers 20K12068 to Yoshio Nakano and JP18K11536 to Kenji Yamagishi, and the Nihon University Research Grant for Social Implementation to Kenji Yamagishi. The funders had no role in study design, data collection and analysis, decision to publish, or preparation of the manuscript.

**Competing interests:** The authors have declared that no competing interests exist.

*Yersinia* spp. [3]. Since these candidate genes are not ubiquitously present among different bacterial species, the selection of an appropriate candidate gene is challenging. Furthermore, phylogenetic classification based on a single gene sequence is not reliable because the selected gene may have undergone horizontal transfer among bacterial species, thus not accurately reflecting the background or history or evolutionary of the species.

To avoid these limitations of phylogenetic analysis based on a single gene, multi-locus sequence analysis/typing (MLSA/MLST) or core genome multi-locus sequence typing (cgMLST) is applied. For *Yersinia*, a genotyping strategy based on five genes was first developed in 2005 [4], and a seven-gene MLST scheme was established to differentiate among the three human pathogenic species (*Y. pestis*, *Y. pseudotuberculosis*, and *Y. enterocolitica* [5]). However, these MLST systems were not applicable at the species level. A genus-wide seven-gene MLST [6] and core-genome MLST [7] were developed for phylogenetic analysis of *Yersinia* species, and demonstrated notably better resolution and phylogenetic precision than the previous MLST methods. The core-genome MLST scheme facilitates high-resolution and efficient identification of various pathogens [8–12]. MLST systems with only a few genes do not reflect the evolutionary history of whole genomes. Species-specific, fixed sets of conserved genes throughout the genome can potentially be used by cgMLST.

Among the aforementioned examples, the evolutionary history of *Yersinia* spp. is poorly understood. *Yersinia* is a Gram-negative bacterium which includes 19 species, 3 being prominent human pathogens: *Y. pestis*, *Y. pseudotuberculosis*, and *Y. enterocolitica* [13]. Various virulence genes from Yersinia species have been reported, and phylogenetic analyses have been performed based on the sequence homology of specific virulence sequences selected from those genes. In this study, we aimed to develop a method to phylogenetically assess any species or subspecies, including non-virulent strains, independent of specific gene sequence homology to characterized species.

The K-mer frequency in DNA fragments provides a genome-specific parameter for analyzing genome diversity [14, 15]. Self-organizing maps (SOMs) based on K-mer nucleotide frequency analysis are used to cluster and visualize DNA fragments derived from closely related eukaryotes [16] or prokaryotes [17] in environmental samples. Sims et al. (2009) reported an alignment-free method, including the feature (or K-mer) frequency profiles (FFPs) of whole genomes [18], and a phylogenetic tree of 10 mammalian species was constructed from intron genomes based on 18-mer frequency profiles. The resultant tree closely correlated with the accepted evolutionary history. Furthermore, they reported a whole-genome-based phylogenetic tree of *E. coli* and *Shigella*, which was constructed using 24-mer frequency profiles [19]. All possible combinations of the 24-mer fragments produced ca 8.4 million features. In addition, various reports on alignment-free sequence comparisons have been published [20]. However this method does not seem to be widely used to date. The authors have created a web application, Phy5, as well as its command line version Phy5cli, which allow for easy phylogenetic tree analysis using this method.

In this study, we constructed an web program for phylogenetic analysis based on 512 pentamer frequencies. Here, we describe its application using the phylogenetic separation of *E. coli*, *Shigella*, *Yersinia*, *Klebsiella*, and *Neisseria* spp. as examples, as well as its command line version for standalone use to generate distance matrix and newick tree files.

A support vector machine (SVM) is a supervised machine-learning model and is one of the most recently developed types of classifiers. It segregates classes using hyperplanes generated by mapping predictors into a new, higher-dimensional space (the feature space), wherein they can be linearly segregated. We hypothesized that SVM would help classify numerous samples of species based on K-mer frequencies from genetic sequences in a short period, and this application is verified here.

## Materials and methods

### Genome sequences

In total, 110 *Yersinia*, 888 *E. coli*, 92 *Shigella*, 280 *Campylobacter*, 561 *Klebsiella*, 67 *Listeria*, 188 *Neiserria*, and 18 *E. albertii* genome sequences were obtained from the ftp site at NCBI Microbial Genomes (ftp://ftp.ncbi.nlm.nih.gov/genomes/Bacteria/). Some strains contained ≥ two genomes or megaplasmids. Genomes or plasmids with molecular size more than one-tenth the size of the largest genome or plasmid were combined into one nucleotide sequence, and those with size less than one-tenth the size of the largest genome or plasmid were excluded.

The chloroplast genomes of *Ipomoea* species were obtained from the ftp site at NCBI plastid (https://ftp.ncbi.nlm.nih.gov/refseq/release/plastid/).

### K-mer frequencies and phylogenetic trees

The tri-, tetra-, penta-, and hexanucleotide frequencies in the genomes of each bacterial sample were determined using R 4.12 (http://www.r-project.org) with the Biostrings package. Degenerated frequencies of K-mer nucleotides, in which complementary K-mer pairs (e.g., AAA vs. TTT) were considered as part of the same nucleotide string, were used to construct phylogenetic trees via hierarchal cluster analysis with Manhattan distance and Ward's algorithm.

### Phylogenetic tree based on MLST

A phylogenetic tree was drawn using multilocus sequence typing (MLST) analysis as described by Duan, et al [5]. The housekeeping genes, *adk* (adenylate kinase), *argA* (amino acid acetyltransferase), *aroA* (3-phosphoshikimate 1-carboxyvinyltransferase), *glnA* (glutamine synthetase), *thrA* (aspartokinase-homoserine dehydrogenase I), *tmk* (thymidilate kinase), and *trpE* (anthranilate synthase component I), were picked from the aforementioned genome sequences. A dendrogram was generated from the concatenated gene sequences via the neighbor joining (NJ) method.

### Phylogenetic tree based on 16S rRNA gene sequences

We extracted 16S rDNA sequences from the aforementioned genome sequences and aligned them using MAFFT [21]. These sequences were used to construct neighbor-joining trees using MUSCLE [22].

### Machine learning

Analysis and classification of the bacterial genome of each strain were accomplished using R 4.12 and the e1071 packages for the SVM with the radial basis function (RBF). The radial kernel function transformed the data using the non-linear function $k(x1, x2) = exp(\gamma|x1 - x2|2)$, where $\gamma$ determines the RBF width, unless specified otherwise. Classification using machine learning was performed via the leave-one-out cross-validation method, i.e., each sample was classified through supervised machine learning using the other 947 samples.

### R web application

Phy5 was coded in R and Bioconductor using the shiny, biostrings, ape, and pvclust packages. These can be further customized or extended using HTML and CSS as Shiny applications. The distance method can be utilized via one of the following: 'euclidean','manhattan', 'maximum',

'canberra', and 'binary'. The agglomeration method can be utilized via one of the following: 'ward', 'average', 'single', 'complete', 'mcquitty', 'median', and 'centroid'.

## Command line application

Phy5cli is a command line version of Phy5. The core portion of Phy5 was embedded in a Python script by rpy2 (https://rpy2.github.io/). It requires Python 3 with the numpy, pandas, and rpy2 libraries, in addition to R (version ≥ 4.0) with the importr, ocalconverter, py2rpy, rpy2py, pandas2ri, numpy2ri, Biostrings, ctc, ape, and pvclust packages. Distance and agglomeration methods can performed in the same manner as through the R web application. The command line version can treat one fasta file containing nucleotide sequences of each strain or species, or multiple fasta files in one directory, which contains nucleotide sequences of fragments from one strain or species. Both versions of Phy5 are available on the git repository github.com/YoshioNakano2021/phy5.

## Results and discussion

Phylogenetic analysis based on 16S rRNA gene sequences is not suitable for closely related species. A bacterial 16S rRNA gene sequence contains 9 variable regions, and the 500-nucleotide long hypervariable V1–V3 region or the 800-nucleotide long V1–V4 region are used for phylogenetic analysis. In closely related species, these variable regions do not vary sufficiently for accurate analyses. *E. coli*, *Salmonella*, *Shigella*, *Yersinia*, and *Klebsiella* from the Enterobacteriaceae family share up to 99% sequence homology of their 16S rRNA gene sequences [23]. In many cases, bacterial operational taxonomic units (OTUs) are defined as 97% homology, thus, species sharing 99% rRNA gene sequence homology cannot be differentiated from one another using 16S rRNA gene sequencing. In addition, prokaryotic species have two types of rRNA operons, and horizontal gene transfer is observed among many bacterial species [24]. The genetic interoperability and promiscuity of 16S rRNA in the ribosomes of an extremely thermophilic bacterium, *Thermus thermophilus*, has been reported by Miyazaki et. al. (2019); they suggested that horizontal gene transfer promotes adaptive evolution, and proposed the "random patch model" for ribosomal evolution [25].

Fig 1 shows the scheme of the alignment-free method for the phylogenetic analysis of closely related species that are resistant to multicopy, horizontal transfer, or chimera-formation of the 16S rRNA genes. This method does not require species-specific gene sets, such as MLSA/MLST, and will allow rapid and simple phylogenetic analysis of any species. For example, in the context of attempting interspecies crossing of morning glory without being familiar with the genus, *Ipomoea*, the proposed method can provide rapid and useful information about the phylogenetic relations between a collection of species within the genus to study its genetic characteristics via MLSA/MLST. Whole-genome comparison is required if specific genes are not selected for phylogenetic analysis, and such a comparison would limit inconsistencies due to gene transfer, multicopy, or chimera-formation of the 16S rRNA genes. As mentioned above, SOM based on K-mer nucleotide frequency analysis has been reported for environmental samples [16, 17]. Frequency analysis of short nucleotide sequences is applicable for any genus or species, as it does not require the selection of specific genes for phylogenetic analysis, as shown in Fig 1. In addition, complete genome sequences are not required for this method, and accumulations of short nucleotide sequences, such as the results of NGS, are sufficient for the analysis. Based on this concept, we demonstrate the efficiency and practicality of phylogenetic analysis using pentanucleotide frequencies.

Degenerated pentanucleotide frequencies were calculated from the DNA sequences of 110 *Yersinia* genomes (S1 Data), and phylogenetic analyses were performed according to the

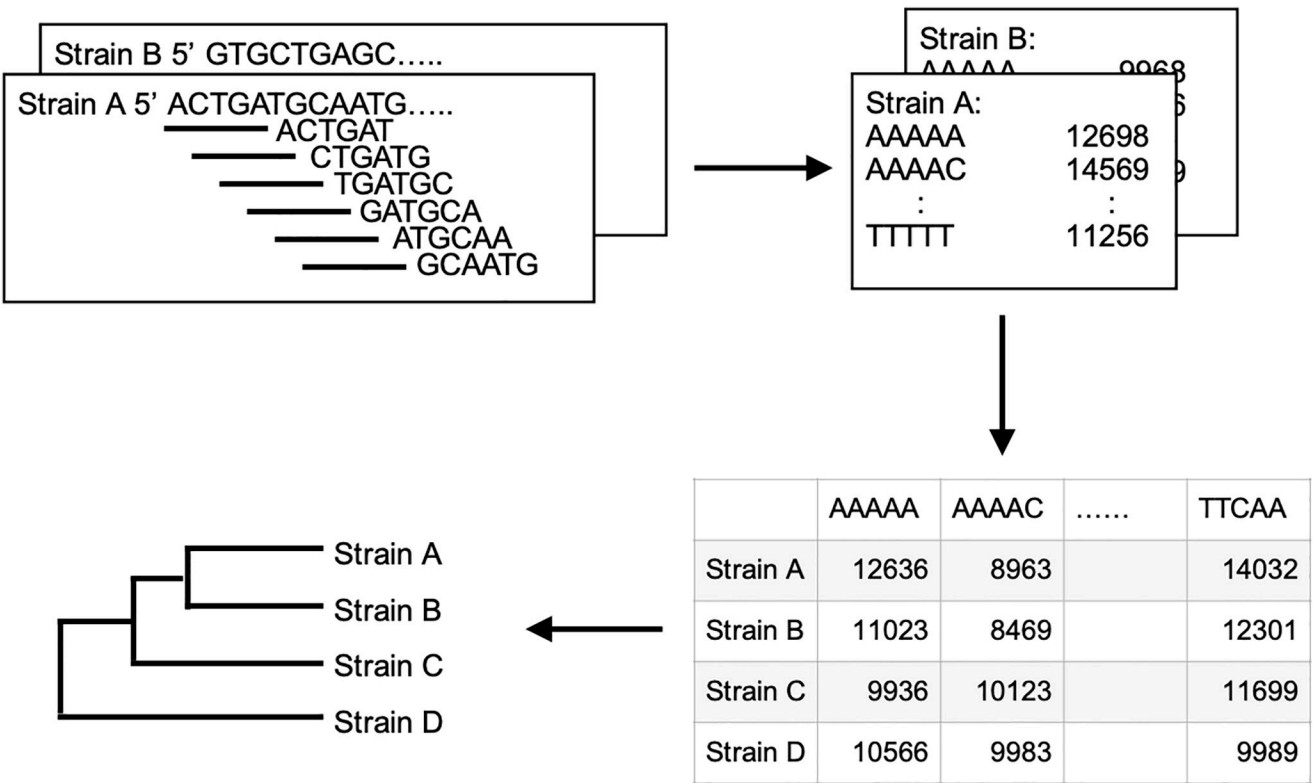

**Fig 1. Scheme for constructing a phylogenetic tree based on pentanucleotide frequencies using Phy5.**

results (Fig 2A). The phylogenetic tree, based on pentamer frequencies with 100% bootstrap values, demonstrated that *Y. pestis*, *Y. pseudotuberculosis*, *Y. enterocolitica*, and *Y. ruckeri* were clearly distinct from one another, as shown in Fig 2A. This correlated with the MSLT (Fig 2B), except that 5 strains in *Y. pestis* and *Y. pseudotuberculosis* were separated from the major strains based on MLST analysis. Further examination might provide more suitable housekeeping genes for MLST. This result suggests that this method (using pentanucleotide frequencies) produces almost identical results to MLST, which has been reported to be a superior analysis method, but is based on a completely different approach with far simpler methodology. On the other hand, 16S rRNA gene sequence-based analyses (S1 Fig) could not distinguish between these species. Furthermore, phylogenetic trees based on tri-, tetra-, penta-, and hexa-nucleotide frequencies (S1–S4 Data) were constructed (S2 Fig). The latter three trees distinguished between three species of *Yersinia* with 100% bootstrap values. Hexamer frequencies distinguished between all strains with $\geq$ 99% bootstrap values. However, this result does not necessarily determine the minimum number of nucleotide frequencies required for phylogenetic tree analysis. Furthermore, this number cannot be calculated prospectively because the diversity in DNA sequences is not consistent across different species and genera. However, pentanucleotide frequencies are sufficient for the phylogenetic analysis of *Yersinia* and other species described in this study.

Furthermore, the phylogenetic trees for *E. coli/Shigella* (Fig 3, S3 Fig), *Campylobacter* (S4 Fig), *Klebsiella* (S5 Fig), *Neisseria* (S6 Fig), and *Escherichia albertii*, including *E. coli* and *Shigella* (S7 Fig) species/strains were generated using pentanucleotide frequencies (S1 Data).

*E. coli* and *Shigella* are difficult to distinguish between based on their 16S rDNA sequences alone, and numerous alternatives have been proposed for their phylogenetic analysis [1, 2, 26].

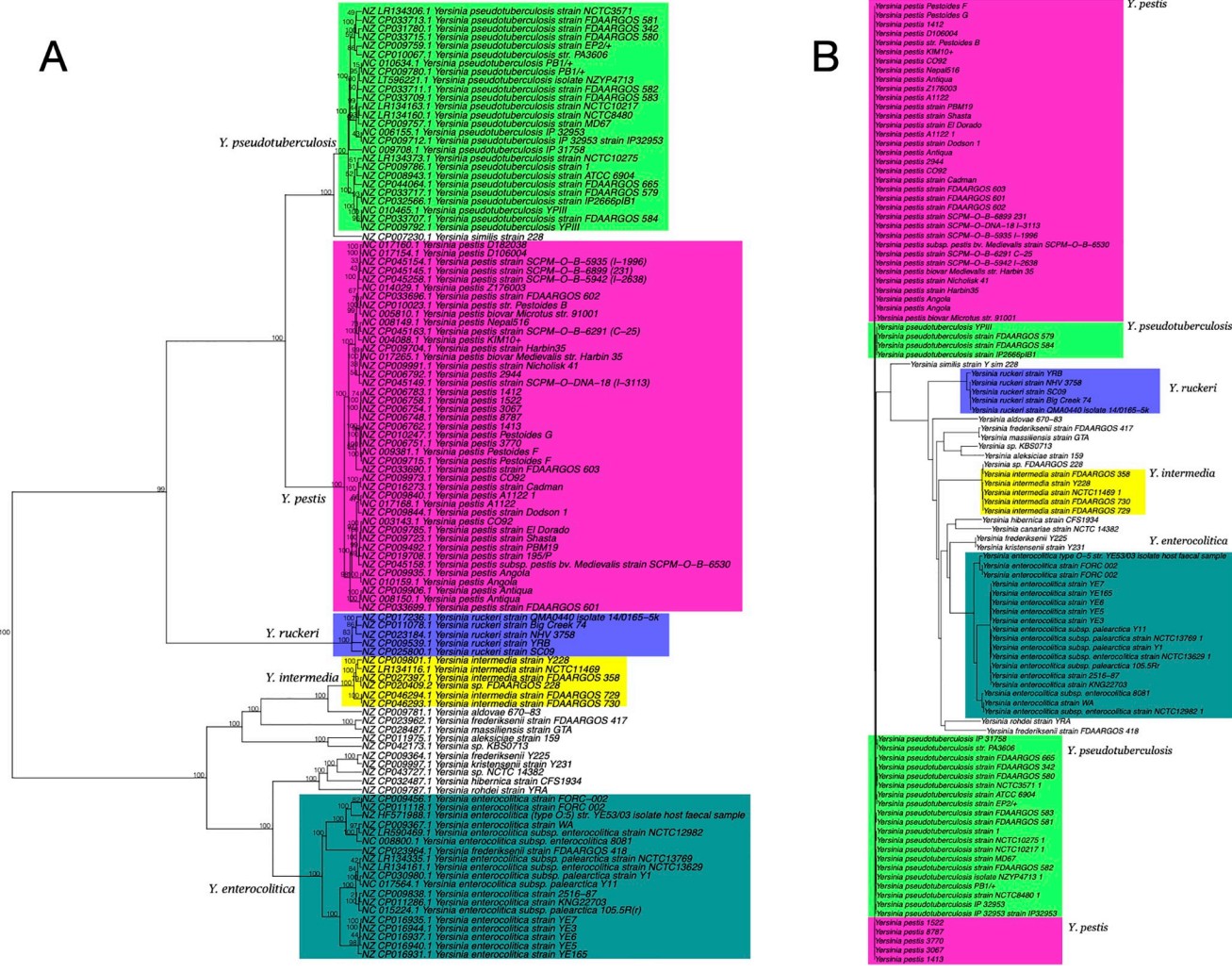

**Fig 2. Phylogenetic trees of 110 Yersinia strains based on pentanucleotide frequencies (A) and MLST analysis (B).** The trees were constructed using the Manhattan distance and Ward's algorithm (A) or the neighbor-joining method (B). The numbers at the nodes indicate the percentage occurrences among 1,000 bootstrap values. Separated groups of species are highlighted.

For example, Fukushima et al. reported phylogenetic analyses of *Salmonella*, *Shigella*, and *E. coli* strains based on sequences of the *gyrB* gene [1], whereas Escobar-Páramo et al. used four chromosomal genes (*trpA*, *trpB*, *pabB*, and *putP*), or three plasmid genes (*ipaB*, *ipaD*, and *icsA*) [2]. Phylogenetic trees based on pentanucleotide frequencies for the four species of *Shigella* and *E. coli* strains are distinct (S3 Fig), and bootstrap values indicate the significance of the separation.

*E. albertii* was originally classified as a *Hafnia alvei*-like strain isolated from human stool samples in the early 1990s, and was suspected of causing diarrhea [27]. It has recently been recognized as a close relative of *E. coli*. Further, *E. albertii* strains were found to be closely related to strains of *Shigella boydii* serotype 13, a distant relative of *E. coli*, representing a divergent lineage in the *Escherichia* genus [28]. Furthermore, Hyma et al. reported that the *E. albertii*-*Shigella* B13 lineage is estimated to have diverged from an *E. coli*-like ancestor 28 million years ago. Herein, we constructed the phylogenetic tree of *E. albertii*, *E. coli*, and *Shigella* from

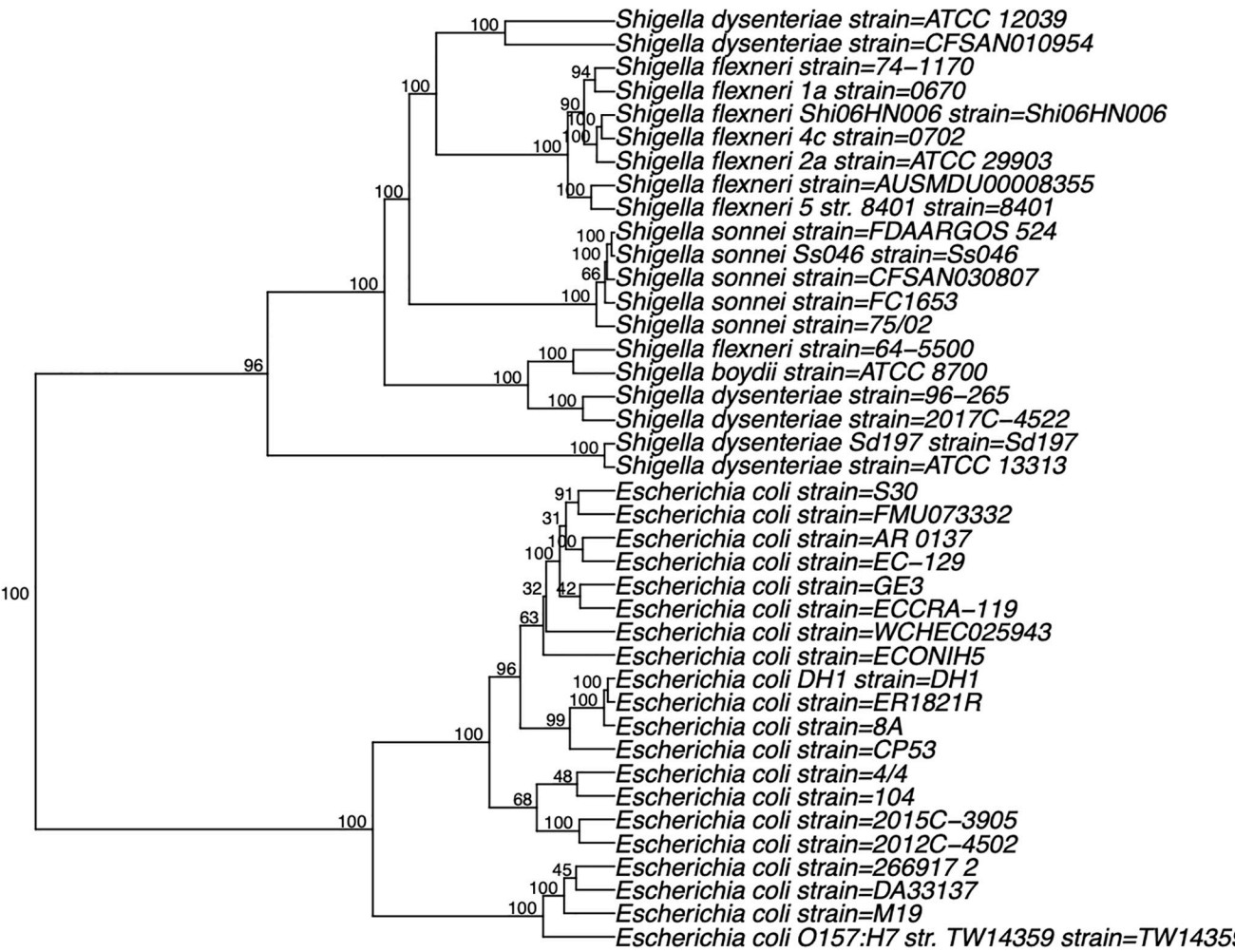

**Fig 3. Phylogenetic trees of 20 random sequences from 888 sequences of *E. coli* and 20 random sequences from 92 sequences of *Shigella* based on pentanucleotide frequencies.**

pentanucleotide frequencies (Fig 4). Fig 4 shows the phylogenetic relationships of lineages of the three species. *E. albertii* and enterohemorrhagic *E. coli* strains, including O157, O121, and O111, are closely related and are on separate branches distinct from *Shigella* and nonpathogenic *E. coli*, such as strain K-12. Fig 4 shows that *Shigella* strains are distinguishable from enteroinvasive *E. coli* (EIEC), and these groups form a large branch separate from the other *E. coli* strains, which includes K12. Three clusters within *Shigella* appear to have diversified over a period of 35,000–270,000 years [26]. One *E. coli* and two *Shigella* species were identified in opposite genera (S3 Fig). *E. coli* NCTC11104 was classified as *Shigella*, whereas it is now identified as Citrobacter Sc16. *S. flexneri* C32 and *Shigella* sp. PAMC 28760 were classified as *E. coli*. The former bacterium was isolated from a Himantormia sp. lichen in Antarctica; thus, its evolutionary history could be different from those of other *Shigella* spp.

Sims and Kim (2011) reported the whole-genome phylogeny of the *E. coli/Shigella* group based on FFPs [19]. The phylogenetic tree was constructed using all possible features of 24-nucleotide segments, which constituted approximately 8.4 million sequences. The feature compositions were refined to core features of ca 0.56 million, with low frequency and low

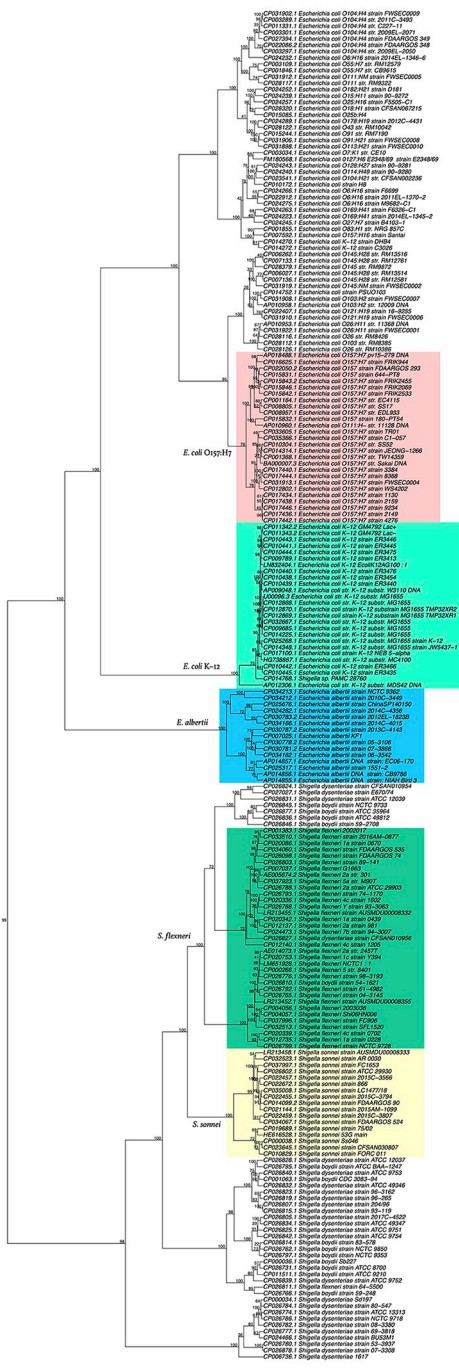

**Fig 4. Phylogenetic trees of *E. albertii*, *Shigella*, and *E. coli* with serotype information based on their pentanucleotide frequencies.** The numbers at the nodes indicate the percentage occurrences among 1,000 bootstrap values. Separated groups of species and serotypes are highlighted.

variability. S3 Fig shows that 512 features of pentanucleotides are adequate to construct a phylogenetic tree for *E. coli/Shigella*.

This approach can be applied to any species without defining species-specific gene sets. As shown in S4–S6 Figs, *Campylobacter*, *Klebsiella*, and *Neisseria* species were clearly distinguished as separate phylogenetic lineages. The construction of phylogenetic trees using genome sequences takes approximately 10 min. Furthermore, rather than determining the whole genome sequence, this approach only requires clusters of short (100 bp) sequences determined through pyrosequencing.

We built an application for phylogenetic analysis of species based on pentanucleotide frequencies as described above. Phy5 is written in R and is freely available at github.com/YoshioNakano2021/phy5. The online version of Phy5 can be accessed at https://phy5.shinyapps.io/Phy5R/. FASTA files containing genome nucleotide sequences (one file for one strain) are uploaded onto the site, and Phy5 draws the resultant phylogenetic tree and tables of proportional pentanucleotide frequencies using aggregated data. In addition, a command line version of "Phy5" has been constructed to use in bioinformatic pipelines. It supports a set of FASTA-formatted files, treating each file as a sample, and a single FASTA-formatted file, treating each sequence as a sample. In addition, a file of the resultant distance matrix and a newick format tree file are generated and saved. An R web application is not suitable for command line applications, and so this version is embedded in Python. It is available at https://github.com/YoshioNakano2021/phy5.

One of the authors (Y.N.) attempted to create interspecies hybrids among *Ipomoea* species, including morning glories, and used a phylogenetic tree for choice combinations of species. Nucleotide sequences of 30 chloroplast genomes from 24 *Ipomoea* species were downloaded from the GenBank sequence database, and their phylogenetic tree was constructed using the Phy5 system described above (Fig 5). The chloroplast DNAs were 162–165 kb long, and it took only a few seconds to calculate pentanucleotide frequencies and construct the phylogenetic tree. The resultant phylogenetic tree correlated well with previously reported trees when comparing their specific genes and morphological characteristics [29–31]. In addition, *I. purpurea*, *I. nil*, *I. tricolor*, and *I. hederacea* are all morning glories, but crossings among *I. purpurea*, *I. nil*, and *I. hederacea* rarely succeed. Incidentally, *I. quamoclit* and *I. hederifolia* are cypress vines, and *I. batatas* is a sweet potato variety; they cannot successfully be crossed with morning glories.

As shown above, only closely related species are suitable for this phylogenetic analysis method, which is based on pentanucleotide frequencies. Fig 6 shows a phylogenetic tree of various distantly related species, including thermophilic archaea and bacteria. *Pyrococcus* and a *Thermococcus* strain form a clade with the *Streptococcus* species, and these thermophilic archaea are distinct from *Pyrobaculum* and other *Thermococcus* species. The latter thermophilic archaea and thermophilic bacteria, such as *Thermotoga*, fall within the same clade. Long nucleotide frequencies (i.e., 18 or 24 nucleotide sequences) are suitable for the phylogenetic analyses of distantly related species, as reported by Sims, et al. (2009). [18, 19]. The combination $2^{24}$ is tedious to calculate, thus, a limited sequence set must be selected to increase specificity for the target species.

SVM classified 856 strains of *E. coli* and 91 strains of *Shigella*, except one *Shigella* strain with 99.9% accuracy (Table 1).

This study shows that phylogenetic analysis based on pentanucleotide profiles is an alternative method for microbial phylogenetic analyses. Future studies are required to identify K-mer sequence combinations that are more specific to bacterial species and develop more efficient calculation methods for microbial phylogenetic analysis.

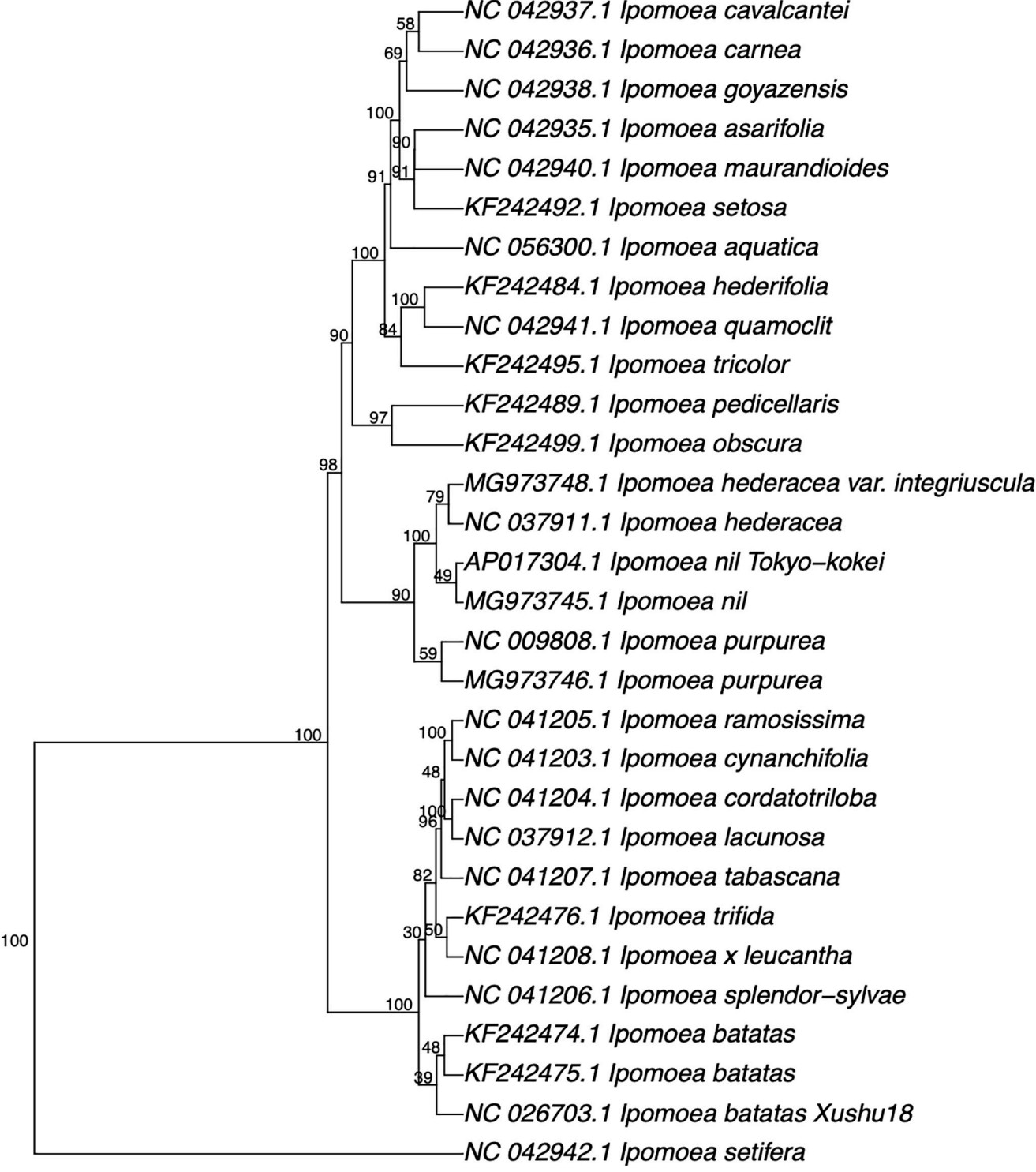

**Fig 5. Phylogenetic trees of *Ipomoea* species based on their chloroplast genomes.** The numbers at the nodes indicate the percentage occurrences among 1,000 bootstrap values. Separated groups of species and serotypes are highlighted.

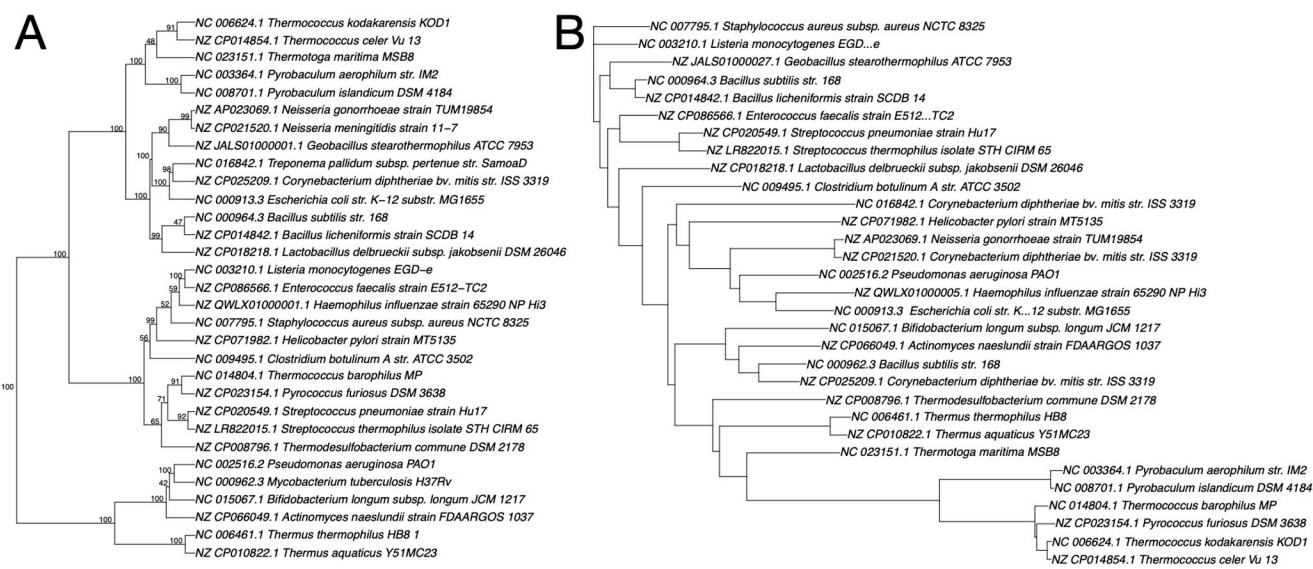

**Fig 6. Phylogenetic trees of various distantly related species including archaea and bacteria, based on pentanucleotide frequencies (A) and 16S rRNA gene sequences (B).** The trees were constructed using the Manhattan distance and Ward's algorithm (A) or the neighbor-joining method (B). The numbers at the nodes indicate the percentage occurrences among 1,000 bootstrap values. Separated groups of species are highlighted.

**Table 1. Classification of *E. coli* and *Shigella* by SVM.**

| Classfication | *E. coli* | *Shigella* |
|---|---|---|
| *E. coli* | 856 | 1 |
| *Shigella* | 0 | 91 |

# Supporting information

**S1 Fig. Phylogenetic trees of 110 Yersinia strains based on pentanucleotide frequencies (A) and 16S rRNA gene sequences (B).** The trees were constructed using the Manhattan distance and Ward's algorithm (A) or the neighbor-joining method (B). The numbers at the nodes indicate the percentage occurrences among 1,000 bootstrap values. Separated groups of species are highlighted.
(TIFF)

**S2 Fig. Phylogenetic trees based on (A) tri-, (B) tetra-, (C) penta-, and (D)hexa- nucleotide frequencies (S2–S4 Data) were constructed.** The trees were constructed using the Euclidean distance and Ward's algorithm. The numbers at the nodes indicate the percentage occurrences among 1,000 bootstrap values.
(TIFF)

**S3 Fig. Phylogenetic trees for *E. coli/Shigella* based on pentanucleotide frequencies.** The trees were constructed using the Euclidean distance and Ward's algorithm. The numbers at the nodes indicate the percentage occurrences among 1,000 bootstrap values.
(TIFF)

**S4 Fig. Phylogenetic trees of *Campylobacter* species based on pentanucleotide frequencies.** The trees were constructed using the Euclidean distance and Ward's algorithm. The numbers at the nodes indicate the percentage occurrences among 1,000 bootstrap values.
(TIFF)

**S5 Fig. Phylogenetic trees of *Klebsiella* species based on pentanucleotide frequencies.** The trees were constructed using the Euclidean distance and Ward's algorithm. The numbers at the nodes indicate the percentage occurrences among 1,000 bootstrap values.
(TIFF)

**S6 Fig. Phylogenetic trees of *Neisseria* species based on pentanucleotide frequencies.** The trees were constructed using the Euclidean distance and Ward's algorithm. The numbers at the nodes indicate the percentage occurrences among 1,000 bootstrap values.
(TIFF)

**S7 Fig. Phylogenetic trees of *Escherichia albertii*, including *E. coli* and *Shigella* species based on pentanucleotide frequencies.** The trees were constructed using the Euclidean distance and Ward's algorithm. The numbers at the nodes indicate the percentage occurrences among 1,000 bootstrap values.
(TIFF)

**S1 Data. CSV file containing the degenerated pentanucleotide frequencies in the DNA sequences of *Yersinia*, *E. coli*, *Shigella*, *Campylobacter*, *Klebsiella*, *Listeria*, *Neisseria*, and *E. albertii*.**
(CSV)

**S2 Data. CSV file containing the degenerated trinucleotide frequencies in the DNA sequences of *Yersinia* species.**
(CSV)

**S3 Data. CSV file containing the degenerated tetranucleotide frequencies in the DNA sequences of *Yersinia* species.**
(CSV)

**S4 Data. CSV file containing the degenerated hexanucleotide frequencies in the DNA sequences of *Yersinia* species.**
(CSV)

## Acknowledgments

We would like to thank Editage (www.editage.com) for English language editing.

## Author Contributions

**Funding acquisition:** Yoshio Nakano, Kenji Yamagishi.

**Methodology:** Yusaku Domon, Kenji Yamagishi.

**Software:** Yoshio Nakano, Yusaku Domon.

**Supervision:** Yoshio Nakano.

**Writing – original draft:** Yoshio Nakano.

**Writing – review & editing:** Yusaku Domon, Kenji Yamagishi.

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
