## [Decision Letter · Decision Letter 0]

27 Oct 2022

PONE-D-22-13256Phylogenetic trees of closely related bacterial species and subspecies based on frequencies of short nucleotide sequencesPLOS ONE

Dear Dr. Nakano,

Thank you for submitting your manuscript to PLOS ONE. After careful consideration, we feel that it has merit but does not fully meet PLOS ONE’s publication criteria as it currently stands. Therefore, we invite you to submit a revised version of the manuscript that addresses the points raised during the review process.

In particular, in order to improve the reader perception on the merit of the new method of phylogenetic recostruction, authors need to provide : 1) a comprehensive comparison with other methods, 2) complete documentation on the R package "Phy5", 3)  clear discussion on potential applications

We look forward to receiving your revised manuscript.

Kind regards,

Frédérique Pasquali

Academic Editor

PLOS ONE

Reviewers' comments:

Reviewer's Responses to Questions

**Comments to the Author**

1. Is the manuscript technically sound, and do the data support the conclusions?

Reviewer #1: Yes

Reviewer #2: Partly

2. Has the statistical analysis been performed appropriately and rigorously? 

Reviewer #1: Yes

Reviewer #2: No

3. Have the authors made all data underlying the findings in their manuscript fully available?

Reviewer #1: Yes

Reviewer #2: Yes

4. Is the manuscript presented in an intelligible fashion and written in standard English?

Reviewer #1: Yes

Reviewer #2: Yes

5. Review Comments to the Author

Reviewer #1: In this manuscript, author presented a useful alternative for phylogenetic analysis of closely related bacterial species. Escherichia coli, Shigella, Yersinia, Klebsiella, and Neisseria spp. or serotypes of Listeria monocytogenes were analyzed, and good results were observed. I think that this method is worth exploring further. This manuscript can be accepted after minor revision.

Minor comment:

(i) The paragraph (lines 153-161) is a repetition of the previous one (lines 144-152).

(ii) Supplementary Data 2-4 is missing from the manuscript.

(iii) Line 162: stain→strain.

(iv) Line 174: E. coli should be in italics.

(v) Line 210: Please revise ‘textitI.’

Reviewer #2: Review of the manuscript PONE-D-22-13256, titled "Phylogenetic trees of closely related bacterial species and subspecies based on frequencies of short nucleotide sequences".

The article describes a method of phylogenetic reconstruction based on pentanucleotide frequencies. The method could be potentially useful. Unfortunately the current manuscript does not provide a comprehensive comparison with other methods, which makes it difficult to judge the merit of the new method. Also, the implementation of the method needs a documentation and a command line interface to be useful in practical applications. I think, this could be a useful contribution, if these points are addressed.

Main comments:

1. Fig.2 compares trees based on pentanucleotide frequencies and 16S rRNA sequences. However, as you noted in the introduction, MLST methods have been already proposed for distinguishing Yersinia species. Such MLST-based tree, based on alignment of concatenated multiple gene sequences, should be included in this comparison. You should compare your method to available state-of-the art alternatives, not just to the weakest alternative method.

2. Figs 3, 4, 5 and 6 show trees based on pentanucleotide frequencies. Best known alignment-based methods should be shown in comparison. This would allow judging the accuracy of the proposed new method.

3. The manuscript needs a clear discussion about scenarios where the new method based on pentanucleotide frequencies should be preferred to other methods, and in which cases it should not be used.

4. The new method is implemented in an R package "Phy5". The package lacks documentation. There should be a step-by-step manual about how to install it, and apply it to a FASTA-formatted dataset. Ideally two scenarios should be supported: (1) Operating on a set of FASTA-formatted files, treating each file as a sample. (2) Operating on a single FASTA-formatted file, treating each sequence as a sample. For this method to be useful in practice, there should be a non-interactive way to generate distance matrix (e.g., in PHYLIP format) and a tree (in newick format), from command line.

5. I think the introduction part does not adequately introduce the topic of alignment-free sequence comparison, and the previous related works. It should include more examples of similar work, because a lot of studies have been done on this topic. See, e.g., these resources for some, but there are many more:

https://genomebiology.biomedcentral.com/articles/10.1186/s13059-017-1319-7

https://en.wikipedia.org/wiki/Alignment-free_sequence_analysis

6. It's not clear what is the unique or novel point of this work. Sequence comparison based on nucleotide frequencies have been studies a lot in the past. A possible contribution of this article could be, e.g., a comprehensive comparison of this method with other methods, and a guide about when it's applicable. Or, a convenient implementation, allowing easy reuse of this method in practical bioinformatic pipelines. But currently the manuscript is also lacking in both of these regards.

Minor comments.

7. Text, Line 28: "Yersinia is a gram-negative bacterium consisting of 19 species and includes 3 prominent human pathogens: Y. pestis, Y. pseudotuberculosis, and Y. enterocolitica [13]." ¬- I think more than 19 species are currently recognized, e.g., see the NCBI Taxonomy Database: https://www.ncbi.nlm.nih.gov/Taxonomy/Browser/wwwtax.cgi?mode=Tree&id=629&lvl=2&lin=f&keep=1&srchmode=1&unlock

8. Text, Line 153: This part repeats the previous paragraph.

6. PLOS authors have the option to publish the peer review history of their article (what does this mean?). If published, this will include your full peer review and any attached files.

Reviewer #1: No

Reviewer #2: No

---

## [Author Response · Author response to Decision Letter 0]

8 Mar 2023

Reviewer 1:

Reviewer:

Minor comment:

(i) The paragraph (lines 153-161) is a repetition of the previous one (lines 144-152).

(ii) Supplementary Data 2-4 is missing from the manuscript.

(iii) Line 162: stain→strain.

(iv) Line 174: E. coli should be in italics.

(v) Line 210: Please revise ‘textitI.’

Authors: 

(i) The duplicated sentences have been removed. 

(ii) Supplementary Data 2-4 will be surely uploaded on the revised manuscript submission. 

(iii) The typo has been corrected. 

(iv) The word, E. coli, has been italicized. 

(v) The incomplete LaTeX command has been corrected. 

Reviewer 2:

Main comments:

Reviewer:

1. Fig.2 compares trees based on pentanucleotide frequencies and 16S rRNA sequences. However, as you noted in the introduction, MLST methods have been already proposed for distinguishing Yersinia species. Such MLST-based tree, based on alignment of concatenated multiple gene sequences, should be included in this comparison. You should compare your method to available state-of-the art alternatives, not just to the weakest alternative method.

Authors: 

Figure 2B has been replaced with a phylogenetic tree based on the MLST method, and the 16S rRNA-based tree has been renamed as Supplementary Figure 1. The trees generated using pentanucleotide frequencies and MLST have very similar structures.

Reviewer:

2. Figs 3, 4, 5 and 6 show trees based on pentanucleotide frequencies. Best known alignment-based methods should be shown in comparison. This would allow judging the accuracy of the proposed new method.

Authors: 

The MLST method for Yersinia has been reported and we can analyze it according to the previously published papers. In contrast, effective MLST methods for Escherichia coli and Shigella have not been reported, so candidate genes for MLST analysis must be determined. If the reviewer believes that MLST analysis for E. coli and Shigella for Figure 4 and Figure 3 are indispensable for this paper, we will attempt such MLST analysis. However, this will likely take around 6 months to complete based on the timeline of the Japanese academic year.

MLST or phylogenetic analyses for Ipomoe chloroplasts have also not been reported, so Figure 5 cannot be compared to other alignment-based methods.

A phylogenetic tree built using 16S rRNA gene analysis has been added to Figure 6 according to the reviewer's suggestion.

Reviewer:

3. The manuscript needs a clear discussion about scenarios where the new method based on pentanucleotide frequencies should be preferred to other methods, and in which cases it should not be used.

Authors: 

This application is useful for analyzing bacterial species whose genome sequences have not yet been determined. Accumulation of short sequences can be used for phylogenetic tree analysis without identification of genes in those fragments. This method is suitable for use as a first candidate, as it can be analyzed without the need for prior knowledge of the genes. In addition, it would be difficult to use other methods in combination with SVM to distribute a large number of samples as draft sequence data.

Furthermore, other methods would be difficult to identify species based on a large amount of draft sequence data in combination with SVM. These descriptions have been added at the end of the manuscript.

Figure 6 demonstrates a case where this method should not be used. 

Reviewer:

4. The new method is implemented in an R package "Phy5". The package lacks documentation. There should be a step-by-step manual about how to install it, and apply it to a FASTA-formatted dataset. Ideally two scenarios should be supported: (1) Operating on a set of FASTA-formatted files, treating each file as a sample. (2) Operating on a single FASTA-formatted file, treating each sequence as a sample. For this method to be useful in practice, there should be a non-interactive way to generate distance matrix (e.g., in PHYLIP format) and a tree (in newick format), from command line.

Authors: 

"Phy5" is not an R package. It is a web application based on the R package "Shiny". It can be used as a standalone application and its manual would be the same as that for Shiny.

A command line version of "Phy5" has been constructed to use in bioinformatic pipelines. The command line version supports a set of FASTA-formatted files or a single FASTA-formatted file. In addition, a file of the resultant distance matrix and a newick format tree file are generated and saved.

R scripts are not very useful for command-line use. In particular, it is not suitable for introducing choices by argv. Therefore, we have made the command line version more user-friendly by integrating it into a Python script.

Reviewer:

5. I think the introduction part does not adequately introduce the topic of alignment-free sequence comparison, and the previous related works. It should include more examples of similar work, because a lot of studies have been done on this topic. See, e.g., these resources for some, but there are many more:

https://genomebiology.biomedcentral.com/articles/10.1186/s13059-017-1319-7

https://en.wikipedia.org/wiki/Alignment-free_sequence_analysis

Authors: Content from the review article by Zielezinski has been added to the introduction, and it has been cited.

Reviewer:

6. It's not clear what is the unique or novel point of this work. Sequence comparison based on nucleotide frequencies have been studies a lot in the past. A possible contribution of this article could be, e.g., a comprehensive comparison of this method with other methods, and a guide about when it's applicable. Or, a convenient implementation, allowing easy reuse of this method in practical bioinformatic pipelines. But currently the manuscript is also lacking in both of these regards.

Authors: 

We have submitted this alignment-free sequence comparison to various journals, and for the first time, we have received constructive comments (one of them demanded proof of the evolution from me). We acknowledge that this is no longer a novel idea, but phylogenetic analysis methods based on the frequencies of short sequences are still not widely used. We hope that another web or command line application may be useful for constructing phylogenetic trees based on genome wide sequences. We absolutely agree that command-line use would be a convenient implementation for practical bioinformatic pipelines, and the command line version of Phy5 has been built and embedded into a Python script.

Minor comments.

Reviewer:

7. Text, Line 28: "Yersinia is a gram-negative bacterium consisting of 19 species and includes 3 prominent human pathogens: Y. pestis, Y. pseudotuberculosis, and Y. enterocolitica [13]." ¬- I think more than 19 species are currently recognized, e.g., see the NCBI Taxonomy Database: https://www.ncbi.nlm.nih.gov/Taxonomy/Browser/wwwtax.cgi?mode=Tree&id=629&lvl=2&lin=f&keep=1&srchmode=1&unlock

Authors: "19 species" has been corrected "26 species".

Reviewer:

8. Text, Line 153: This part repeats the previous paragraph.

Authors: The duplicated sentences have been removed.

---

## [Decision Letter · Decision Letter 1]

3 Apr 2023

Phylogenetic trees of closely related bacterial species and subspecies based on frequencies of short nucleotide sequences

PONE-D-22-13256R1

Dear Dr. Nakano,

We’re pleased to inform you that your manuscript has been judged scientifically suitable for publication and will be formally accepted for publication once it meets all outstanding technical requirements.

Within one week, you’ll receive an e-mail detailing the required amendments. Additionally, please consider comment 2 of reviewer 2 at the bottom of this letter.  When these have been addressed, you’ll receive a formal acceptance letter and your manuscript will be scheduled for publication.

Kind regards,

Frédérique Pasquali

Academic Editor

PLOS ONE

Additional Editor Comments (optional):

Reviewers' comments:

Reviewer's Responses to Questions

**Comments to the Author**

1. If the authors have adequately addressed your comments raised in a previous round of review and you feel that this manuscript is now acceptable for publication, you may indicate that here to bypass the “Comments to the Author” section, enter your conflict of interest statement in the “Confidential to Editor” section, and submit your "Accept" recommendation.

Reviewer #1: All comments have been addressed

Reviewer #2: All comments have been addressed

2. Is the manuscript technically sound, and do the data support the conclusions?

Reviewer #1: (No Response)

Reviewer #2: Yes

3. Has the statistical analysis been performed appropriately and rigorously? 

Reviewer #1: (No Response)

Reviewer #2: Yes

4. Have the authors made all data underlying the findings in their manuscript fully available?

Reviewer #1: (No Response)

Reviewer #2: Yes

5. Is the manuscript presented in an intelligible fashion and written in standard English?

Reviewer #1: (No Response)

Reviewer #2: Yes

6. Review Comments to the Author

Reviewer #1: (No Response)

Reviewer #2: I believe the authors addressed my concerns. This manuscript should be acceptable now. The only minor comment: The legend for Fig 6 should explain what is shown on panels A and B.

7. PLOS authors have the option to publish the peer review history of their article (what does this mean?). If published, this will include your full peer review and any attached files.

Reviewer #1: No

Reviewer #2: No

---

## [Editor Report · Acceptance letter]

12 Apr 2023

PONE-D-22-13256R1 

Phylogenetic trees of closely related bacterial species and subspecies based on frequencies of short nucleotide sequences 

Dear Dr. Nakano:

I'm pleased to inform you that your manuscript has been deemed suitable for publication in PLOS ONE. Congratulations! Your manuscript is now with our production department. 

Kind regards, 

on behalf of

Dr. Frédérique Pasquali 

Academic Editor

PLOS ONE